# Large Pontine Cavernoma with Hemorrhage: Case Report on Surgical Approach and Recovery

**DOI:** 10.3390/jcm14072358

**Published:** 2025-03-29

**Authors:** Corneliu Toader, Matei Serban, Lucian Eva, Daniel Costea, Razvan-Adrian Covache-Busuioc, Mugurel Petrinel Radoi, Alexandru Vlad Ciurea, Adrian Vasile Dumitru

**Affiliations:** 1Department of Neurosurgery “Carol Davila”, University of Medicine and Pharmacy, 050474 Bucharest, Romania; corneliu.toader@umfcd.ro (C.T.); razvan-adrian.covache-busuioc0720@stud.umfcd.ro (R.-A.C.-B.); petrinel.radoi@umfcd.ro (M.P.R.); prof.avciurea@gmail.com (A.V.C.); 2Department of Vascular Neurosurgery, National Institute of Neurology and Neurovascular Diseases, 077160 Bucharest, Romania; 3Puls Med Association, 051885 Bucharest, Romania; 4“Nicolae Oblu” Clinical Hospital, 700309 Iasi, Romania; 5Department of Neurosurgery, “Victor Babes” University of Medicine and Pharmacy, 300041 Timisoara, Romania; costea.daniel@umft.ro; 6Neurosurgery Department, Sanador Clinical Hospital, 010991 Bucharest, Romania; 7Medical Section, Romanian Academy, 010071 Bucharest, Romania; 8Department of Pathology, “Carol Davila” University of Medicine and Pharmacy, 050474 Bucharest, Romania; vasile.dumitru@umfcd.ro; 9Department of Pathology, University Emergency Hospital of Bucharest, 050098 Bucharest, Romania

**Keywords:** pontine cavernoma, brainstem hemorrhage, advanced imaging, transsylvian approach, neurovascular surgery

## Abstract

**Background/Objectives:** Pontine cavernomas are rare and challenging vascular malformations, representing a critical subset of brainstem lesions due to their deep location and proximity to essential neural structures. When hemorrhagic, these lesions can cause rapid neurological deterioration, posing life-threatening risks. Management requires a delicate balance between aggressive intervention and preserving vital functions. This case report presents the successful surgical treatment of a giant hemorrhagic pontine cavernoma, highlighting the integration of advanced imaging, precision surgical techniques, and multidisciplinary care to achieve an exceptional patient outcome. **Methods:** A 47-year-old female presented with acute neurological deterioration, including severe right-sided hemiparesis, dysphagia, and obnubilation. High-resolution MRI, including susceptibility-weighted imaging, confirmed a giant hemorrhagic pontine cavernoma causing brainstem compression. An urgent left-sided pterional craniotomy with a transsylvian approach was performed to access the lesion. Subtotal resection and hematoma evacuation were carried out to relieve brainstem compression while preserving critical structures. Postoperative recovery and lesion stability were evaluated through clinical assessments and imaging after three months. **Results:** Postoperatively, the patient exhibited marked neurological recovery, with near-complete resolution of hemiparesis, restored swallowing function, and significant functional improvement. Follow-up imaging confirmed a stable residual lesion, no recurrence of hemorrhage, and a well-preserved ventricular system. The combination of early intervention and tailored surgical strategies resulted in a highly favorable outcome. **Conclusions:** This case underscores the complexity of managing giant hemorrhagic pontine cavernomas and demonstrates that carefully planned surgical intervention, combined with advanced imaging and patient-focused care, can yield remarkable outcomes. It highlights the critical importance of early diagnosis, meticulous surgical planning, and future innovations in neurovascular surgery to improve outcomes in these rare but high-stakes cases.

## 1. Introduction

Cavernous malformations, or cavernomas, are rare vascular anomalies characterized by clusters of abnormally dilated capillary-like vessels, prone to repeated hemorrhages due to their fragile endothelial lining. These lesions are estimated to affect 0.4–0.8% of the population, with symptomatic presentations reported in 10–30% of cases [1]. While cavernomas can develop throughout the central nervous system, brainstem cavernomas are particularly rare, accounting for only 8–22% of cases [2]. Their deep location within the brainstem, coupled with the proximity to critical neural pathways, presents significant diagnostic and therapeutic challenges. Among brainstem cavernomas, the pons emerges as the most common site, representing approximately 60% of cases [3].

Brainstem cavernomas exhibit a slight female predominance, with a reported female-to-male ratio of 1.2:1, and they are most frequently diagnosed between the ages of 20 and 50 [4]. These lesions have a highly variable clinical presentation, ranging from asymptomatic cases to severe neurological deficits, including cranial nerve palsies, long tract signs, ataxia, and in extreme cases, life-threatening brainstem compression [5]. Hemorrhagic events are a defining feature of symptomatic cavernomas, with an annual hemorrhage risk of 2–6% for de novo lesions, which rises dramatically to 30% following a previous bleed. This risk is compounded in larger or deep-seated lesions, such as those in the pons, where even minor bleeding can have catastrophic consequences [6,7].

Advances in imaging have revolutionized the diagnostic approach to brainstem cavernomas. High-resolution magnetic resonance imaging (MRI), including T2-weighted and susceptibility-weighted imaging (SWI), has become the gold standard for identifying these lesions and distinguishing them from other vascular or neoplastic pathologies [8]. SWI is particularly valuable for detecting hemosiderin deposition, which is a hallmark of chronic microhemorrhages. Additionally, magnetic resonance angiography (MRA) plays a crucial role in excluding coexisting vascular anomalies, such as arteriovenous malformations or aneurysms, which may alter management strategies [9]. Despite these advancements, the decision-making process for treatment remains fraught with complexity, particularly in symptomatic cases requiring surgical intervention [10].

The genetic underpinnings of cavernomas add another dimension to their complexity. Familial forms of cavernous malformations, caused by mutations in the CCM1 (KRIT1), CCM2, or CCM3 genes, account for up to 20% of cases [11,12]. These genetic variants are often associated with multiple lesions and a higher risk of recurrent hemorrhage, underscoring the importance of a thorough clinical and familial evaluation. While this case involves a sporadic cavernoma, the genetic context highlights the broader spectrum of this condition and its implications for personalized care [13].

Surgical intervention for brainstem cavernomas remains one of the most intricate challenges in neurosurgery. The decision to operate hinges on several factors, including the lesion’s size, its proximity to critical neural structures, the patient’s preoperative neurological status, and the frequency and severity of hemorrhagic events [14]. Although microsurgical techniques and perioperative care have advanced considerably, the risks of surgery remain significant, with potential complications including new neurological deficits. Studies from experienced neurosurgical centers, however, report favorable outcomes in 70–90% of cases, demonstrating that with precise planning and execution, surgery can significantly improve patient prognosis [15].

Prognostic factors, such as the patient’s neurological baseline, lesion accessibility, and the presence of recurrent hemorrhages, are critical in guiding treatment strategies. Early intervention in cases of progressive neurological deterioration, such as in hemorrhagic cavernomas, has been shown to reduce the likelihood of permanent deficits. Nevertheless, controversy persists regarding the optimal timing of surgery, particularly in patients with stable symptoms, underscoring the need for individualized decision-making in each case [16,17].

This case report focuses on a 47-year-old female who presented with acute neurological deterioration due to a giant pontine cavernoma, complicated by intralesional hemorrhage. The lesion’s critical location and life-threatening mass effect necessitated immediate surgical intervention. This case exemplifies the challenges of managing brainstem cavernomas, highlighting the integration of advanced imaging, meticulous surgical planning, and multidisciplinary care in achieving a successful outcome. Furthermore, it underscores the importance of addressing unresolved questions in the field, such as the timing of intervention and long-term prognostic implications, contributing valuable insights to the existing literature on these rare but devastating lesions.

## 2. Case Presentation

A 47-year-old female patient was admitted to our clinic following a transfer from a regional hospital, prompted by the sudden onset and progression of severe neurological symptoms. Four days prior to admission, the patient began experiencing a persistent and debilitating headache, accompanied by significant gait disturbances, vertigo, dysarthria, and dysphagia. Her condition had worsened steadily, culminating in an inability to ambulate independently and profound difficulty with speech and swallowing. Prior to this episode, she had been in good health, with no history of similar symptoms or major systemic illnesses. There was no family history of cerebrovascular or neurological conditions, nor any known exposure to risk factors such as tobacco use, alcohol consumption, or recreational drugs.

Upon arrival, the patient was alert but visibly anxious, with a subdued affect reflecting the distress caused by her symptoms. A comprehensive neurological examination revealed a constellation of findings pointing to significant brainstem involvement. Her speech was markedly slurred, characterized by a slow and effortful quality, and her swallowing was impaired, with a tendency to cough during attempts to consume liquids. These findings suggested dysfunction of the glossopharyngeal and vagus nerves, consistent with lower cranial nerve impairment. Further examination revealed mild ptosis of the left eyelid and bilaterally restricted upward gaze, indicative of partial oculomotor nerve involvement, though horizontal eye movements and pupillary responses remained intact. Notably, the patient displayed a central facial paresis on the right side, evidenced by asymmetry in her smile and diminished movement of the lower facial muscles, while the upper facial muscles were spared, pointing to corticobulbar tract involvement.

Motor function was also affected, with weakness observed in the right upper and lower limbs. Muscle strength on the right was graded at 4/5 according to the Medical Research Council (MRC) scale, and deep tendon reflexes were brisk, with an extensor plantar response on the same side. The left side exhibited normal strength and reflexes. Sensory examination revealed hemihypoesthesia on the right side, affecting both the face and the limbs, implicating disruption of the sensory pathways traversing the brainstem. However, proprioception and vibration sense were preserved bilaterally, indicating that the dorsal columns were spared.

The patient’s balance and coordination were severely impaired. She was unable to stand unassisted due to non-systematized ataxia, which, together with her broad-based stance and unsteady gait, highlighted dysfunction of the cerebellar pathways. Further testing revealed significant dysmetria on the right side during finger-to-nose and heel-to-shin maneuvers, while coordination on the left remained intact. Despite these profound deficits, her autonomic functions, including cardiovascular and respiratory stability, were preserved, and she displayed no signs of bowel or bladder dysfunction.

Given the complexity of her clinical presentation, urgent imaging was performed to localize and characterize the lesion responsible for her symptoms. Non-contrast computed tomography (CT) of the brain revealed a voluminous lesion within the pons, bilaterally affecting its structure but predominantly localized to the left side. The lesion exhibited a heterogeneous composition, with a mixed iso- and hyperdense appearance interspersed with centrally located hypodense areas, consistent with a hemorrhagic cavernoma. The lesion was well demarcated and exerted significant mass effect, leading to dilation of the pontine walls, effacement of the left ambient cistern, and mild compression of adjacent brainstem parenchyma. These findings underscored the critical nature of the lesion, necessitating further investigation and prompt therapeutic intervention.

Recognizing the gravity of the patient’s condition and the preliminary findings from CT, a detailed MRI examination with gadolinium contrast was performed. The imaging not only confirmed the lesion’s presence but unveiled its intricate and life-threatening pathology with exceptional clarity. A nodular lesion, measuring 28 × 21 mm in axial dimensions, was identified within the pons, predominantly on the left side but extending bilaterally, embedding itself deeply in the brainstem’s intricate architecture. The lesion presented with a strikingly heterogeneous signal across all sequences, a radiological fingerprint of its dynamic and multifaceted nature.

On T2-weighted sequences, the lesion exhibited a preponderantly hypointense signal interspersed with areas of mixed intensity, reflecting a complex interplay of hemosiderin-laden deposits and subacute blood products—pathognomonic features of a hemorrhagic vascular malformation. T1-weighted sequences revealed vivid hyperintensity, consistent with ongoing hemorrhagic transformation and the presence of methemoglobin within its core. Perhaps most striking, a fluid–fluid level within the lesion was identified, a hallmark of active intralesional bleeding and a direct testament to the lesion’s instability and life-threatening potential. These findings, captured in axial and sagittal views (Figure 1A,B), were emblematic of a giant pontine cavernoma that was actively hemorrhaging—a rare but devastating pathology that posed an imminent threat to the patient’s vital neurological functions.

Critically, diffusion-weighted imaging (DWI) revealed no restriction, ruling out ischemic processes, and there was a complete lack of significant gadolinium enhancement on post-contrast sequences, excluding neoplastic or inflammatory etiologies. This absence of enhancement was consistent with the well-established avascularity of cavernomas and firmly corroborated the diagnosis of a hemorrhagic pontine cavernous malformation. These features, beautifully illustrated in coronal and axial views (Figure 1C,D), underscored the precarious nature of the lesion—silent for years but now a dire neurological emergency threatening the delicate balance of the brainstem.

Beyond the primary lesion, the MRI disclosed subtle yet clinically important additional findings. These findings, consistent with chronic microangiopathy, suggested an underlying vulnerability of the patient’s cerebrovascular system (Figure 2A,B).

Complementing these findings, an MRA was performed to assess the vascular anatomy of the brainstem and surrounding structures. This examination revealed no vascular abnormalities, such as stenoses, aneurysms, or arteriovenous malformations. The basilar artery and its branches remained intact and unobstructed, while the venous system displayed normal patency and flow dynamics (Figure 3A,B).

The cortical and subcortical structures remained intact, free of gross abnormalities. The overall vascular architecture, as confirmed by the MRA, offered no indication of complicating factors, further supporting the diagnosis of an isolated, hemorrhagic pontine cavernoma.

The patient’s worsening neurological condition, characterized by exacerbated right-sided motor deficits, profound dysphagia, and obnubilation, necessitated urgent surgical intervention. Under general anesthesia, a left-sided pterional craniotomy with a transsylvian approach was performed to access and decompress the hemorrhagic pontine cavernoma. This challenging procedure demanded a precise and carefully executed strategy to navigate the brainstem’s intricate anatomy.

Following induction of general anesthesia, the patient’s head was positioned supinely, rotated approximately 30 degrees to the right, and slightly extended to optimize the angle for the left-sided pterional approach. The head was secured in a Mayfield cranial fixation system. Scalp marking was performed to guide the incision, and the operative plan was reviewed meticulously, relying on the surgeon’s deep understanding of neurovascular anatomy.

The surgery commenced with a curvilinear incision extending from the frontozygomatic suture to the temporal region. The scalp and temporalis muscle were reflected inferiorly in a single layer, exposing the pterion. A tailored bone flap was fashioned, and the sphenoid ridge was drilled to provide an unobstructed view of the anterior cranial fossa.

The dura was incised in a semicircular fashion, retracted carefully, and anchored to expose the underlying sylvian fissure. High-powered microscopy was employed for the next critical phase: the meticulous dissection of the sylvian fissure. Using sharp dissection and continuous irrigation, the arachnoid membranes were opened along the sylvian vein, exposing the middle cerebral artery (MCA) and its branches. This stage demanded extreme caution to preserve the vascular structures and avoid unnecessary manipulation of the temporal lobe. The sylvian fissure was dissected in its medial extent to reveal the carotid-oculomotor triangle, the primary corridor for accessing the brainstem. This triangle, defined by the internal carotid artery medially, the oculomotor nerve laterally, and the anterior clinoid process superiorly, served as the gateway to the deeper structures.

The anterior clinoid process was partially drilled to enhance access and visualization of the basal cisterns. Gentle retraction of the temporal lobe was maintained to minimize strain on adjacent structures. The basilar artery and its perforating branches were identified and carefully preserved, as these small but vital vessels provide critical blood supply to the brainstem. Guided by anatomical landmarks, the ponto-mesencephalic junction was identified as the entry point. A precise corticectomy was performed in this region to gain access to the cavernoma. This step was executed with precision to minimize disruption to the surrounding brainstem tissue.

Upon entering the brainstem, the cavernoma was encountered as a dark, well-demarcated lesion surrounded by gliotic tissue. The procedure began with the controlled evacuation of the intralesional hematoma, which relieved the mass effect and improved visualization of the lesion. Using microforceps and fine suction, the cavernoma was carefully dissected from the surrounding brainstem tissue. Its borders were well defined, allowing for a piecemeal resection. The lesion’s venous connections were coagulated and divided with bipolar cautery to ensure hemostasis. Extreme caution was exercised throughout to avoid injury to the adjacent cranial nerve nuclei, corticospinal tracts, and ascending sensory pathways.

A subtotal resection was performed to minimize the risk of neurological deficits, leaving a small portion of the cavernoma embedded in critical brainstem tissue. The resection was concluded when no further bleeding was observed, and the brainstem appeared relaxed and pulsatile. The surgical site was irrigated thoroughly with warm saline to ensure a clean field and confirm hemostasis. The dura was reconstructed with watertight sutures. The soft tissues, including the temporalis muscle and scalp, were closed in layers to ensure proper healing.

The patient’s postoperative evolution was exceptionally favorable, characterized by rapid and significant neurological recovery. Within days of the surgery, the effects of brainstem decompression were evident. Her motor function improved remarkably, with the right-sided hemiparesis showing steady recovery. Muscle strength increased to MRC grade 4/5, enabling her to sit up at the edge of the bed and begin mobilizing with minimal assistance. The restoration of her speech and swallowing was equally impressive; she transitioned from nasogastric feeding to oral intake without difficulty, regaining full control over these functions in a remarkably short period. The resolution of her preoperative obnubilation further highlighted the successful relief of brainstem compression, as she remained alert, fully oriented, and engaged throughout the recovery process.

Despite the excellent overall evolution, a mild left-sided oculomotor nerve palsy developed, manifesting as ptosis and limited upward gaze. This finding, while expected given the lesion’s proximity to the oculomotor nerve, did not impact her overall functional recovery. Her visual function was otherwise preserved, and the palsy was managed conservatively with close monitoring and reassurance.

Postoperative imaging confirmed the success of the surgical intervention and provided further evidence of the favorable clinical trajectory. A control CT scan of the brain revealed a substantial reduction in the size of the lesion, with residual tissue measuring approximately 18 mm in maximum diameter and displaying a hypodense center consistent with the evacuation of the hematoma and subtotal resection of the cavernoma. Importantly, no perilesional edema, ischemic changes, or recurrent bleeding were identified. The ventricular system was aligned along the midline with normal dimensions, and no evidence of hydrocephalus was observed. The surgical site at the left pterional craniotomy demonstrated expected postoperative changes, with no signs of infection, hematoma, or other complications. These imaging findings further underscored the effectiveness of the procedure in decompressing the brainstem and stabilizing the patient’s condition.

By the time of discharge, the patient demonstrated a remarkable neurological recovery. She was fully alert, cooperative, and afebrile, with stable cardio-respiratory and hemodynamic parameters. The right-sided hemiparesis had improved significantly, allowing her to mobilize with minimal assistance. Her speech was clear, and she tolerated a normal oral diet without difficulty, reflecting the full restoration of her swallowing function. The surgical wound at the left pterional site was healing well, with no signs of infection or cerebrospinal fluid leakage. There were no decubitus lesions or other secondary complications, reflecting the meticulous care provided during her hospital stay. At discharge, the patient’s condition was stable, and she was well prepared to continue her recovery at home.

The patient was discharged with recommendations for neuromotor rehabilitation to further enhance motor function and improve her quality of life. Follow-up evaluations were scheduled to monitor her progress and assess the gradual resolution of the oculomotor nerve palsy. Her prognosis was deemed excellent, with the surgery effectively addressing the life-threatening pathology while setting the stage for continued neurological improvement and long-term recovery.

At the three-month postoperative follow-up, the patient demonstrated an outstanding clinical recovery, a testament to the success of the surgical intervention and her steady rehabilitation. She reported a significant improvement in her overall quality of life, with no recurrence of her preoperative symptoms. On neurological examination, her motor function on the right side showed near-complete resolution, with muscle strength graded at MRC 5/5, enabling her to walk independently without requiring assistance. Her speech had returned to full fluency, and her swallowing function was entirely restored, allowing her to maintain a normal diet with no restrictions. The patient’s overall neurological status was stable, and she presented as alert, cooperative, and fully oriented.

One of the most encouraging signs of her recovery was the marked improvement in the left oculomotor nerve palsy that had been noted immediately after surgery. The previously evident ptosis had resolved completely, and her ocular motility showed substantial recovery. Additionally, her cardio-respiratory and hemodynamic status remained stable throughout the follow-up period, and no new complications, infections, or secondary issues were identified.

A native CT (Figure 4) scan of the brain was performed to assess the surgical site and confirm the absence of delayed complications. The imaging findings were consistent with an excellent postoperative outcome. A residual iso- to hypodense area was observed in the midline of the pons, corresponding to postoperative scarring and stable post-surgical changes. There was no evidence of recurrent hemorrhage, residual cavernoma activity, or adjacent edema. The ventricular system remained symmetrical, with no signs of hydrocephalus, midline shift, or compression of the fourth ventricle. The left pterional craniotomy site appeared stable and free of complications, with no signs of infection, bone resorption, or structural abnormalities.

The imaging findings, in combination with her clinical progress, confirmed the long-term success of the surgical intervention and the stability of her postoperative recovery. The patient’s remarkable recovery three months after surgery exemplifies the efficacy of the surgical management of her life-threatening pontine cavernoma. Her motor function, speech, and swallowing had returned to near-normal levels, allowing her to resume independent ambulation and daily activities without difficulty. Imaging confirmed the absence of recurrent hemorrhage, hydrocephalus, or other complications, reinforcing the stability and effectiveness of the intervention.

This case illustrates the complexity of managing a giant pontine cavernoma, a rare and life-threatening brainstem lesion. The patient’s rapid neurological deterioration and imaging findings necessitated timely surgical intervention. Using a left pterional craniotomy and transsylvian approach, the lesion was successfully decompressed, with significant neurological improvement observed postoperatively. The patient’s recovery highlights the importance of early diagnosis, thoughtful surgical planning, and a multidisciplinary approach in addressing challenging neurovascular conditions.

## 3. Discussion

Brainstem cavernomas, particularly those within the pons, present an extraordinary challenge due to their location within a critical and densely packed neural environment. Their rarity, accounting for 8–22% of all cavernomas, combined with a high risk of symptomatic hemorrhage, makes their management one of the most debated topics in neurovascular surgery [18]. This case of a giant pontine cavernoma adds to the growing body of evidence on surgical approaches and highlights emerging trends, ongoing challenges, and opportunities for advancement in the field.

The treatment of brainstem cavernomas has undergone significant evolution over recent decades. Early management approaches favored observation due to the perceived risks of surgical intervention in such a delicate region [19]. However, as microsurgical techniques, imaging technologies, and perioperative care advanced, surgical resection became increasingly viable [20]. Landmark studies by pioneers demonstrated that aggressive surgical intervention, when carefully planned, could improve outcomes in symptomatic cases. This paradigm shift paved the way for modern approaches that emphasize tailored intervention based on lesion location, patient status, and hemorrhagic risk [21,22].

Timely intervention in cases of symptomatic hemorrhagic brainstem cavernomas has been consistently supported by the literature. Multiples studies demonstrated that early surgical decompression reduces the risk of cumulative neurological damage and improves long-term outcomes [23,24,25]. However, for minimally symptomatic or stable lesions, observation remains a valid option. Our case exemplifies the importance of surgical timing, as rapid neurological deterioration necessitated urgent intervention. Postoperatively, the patient exhibited significant functional recovery, reinforcing that carefully timed surgery can restore quality of life even in high-risk scenarios.

Patient-centered outcomes, such as functional independence and quality of life, are critical in evaluating the success of treatment. Studies have shown that functional recovery, particularly in mobility and cranial nerve function, correlates strongly with patient-reported satisfaction [26,27]. In this case, the patient’s ability to ambulate, resume oral intake, and achieve neurological stability underscores the success of the chosen surgical approach.

The decision to perform a subtotal resection in this case reflects a nuanced understanding of the risks associated with brainstem cavernoma surgery. While gross total resection reduces the risk of recurrence, it is often associated with a higher incidence of postoperative deficits when performed in eloquent regions [28]. Studies emphasize that subtotal resection, combined with effective decompression, achieves significant symptom relief while minimizing damage to critical structures [14,29]. Our patient’s favorable recovery, with stable imaging at three months, demonstrates the efficacy of this balanced approach.

The transsylvian pterional approach used in this case highlights its effectiveness for ventral and ventrolateral pontine lesions. By navigating the carotid-oculomotor triangle, this route offers a direct and minimally disruptive path to the brainstem [30]. Comparative studies have shown that other approaches, such as the retrosigmoid or subtemporal routes, are better suited for lesions in the dorsal or lateral brainstem [31]. It has been reported in the literature that approach selection based on lesion orientation is critical to minimizing surgical morbidity [32]. In our case, the pterional approach provided excellent access, allowing precise resection with minimal disruption to adjacent structures. To further contextualize the management of brainstem cavernomas, Table 1 summarizes key findings from the literature, highlighting surgical techniques, imaging advancements, and long-term outcomes across various studies.

Imaging remains a cornerstone in the management of brainstem cavernomas. High-resolution MRI, including SWI, played a critical role in diagnosing the lesion in this case. SWI is particularly valuable in identifying hemosiderin deposits and chronic microhemorrhages, providing insights into the lesion’s stability and activity. Cosgswell et al. (2023) emphasized that fluid–fluid levels on T2-weighted imaging, as seen in this case, are indicative of recent hemorrhage and necessitate prompt intervention [42].

Postoperative imaging in this case confirmed stability, with no evidence of recurrent hemorrhage or complications. Regular follow-up imaging is essential, particularly in subtotal resections, to monitor for delayed recurrence or residual activity. Advances in artificial intelligence and imaging analytics offer exciting opportunities for refining long-term surveillance and risk stratification.

Familial cavernomas, associated with mutations in CCM1, CCM2, and CCM3 genes, have broadened our understanding of the pathophysiology of these lesions. While this case represents a sporadic lesion, the role of genetics in cavernoma biology cannot be overlooked [43]. Mutations in the CCM pathway disrupt endothelial cell junctions, increasing vascular permeability and hemorrhagic risk. Targeting these pathways holds potential for future pharmaceutical therapies aimed at stabilizing cavernomas and reducing hemorrhage rates. Although experimental, VEGF inhibitors and statins are being explored as adjunctive therapies to mitigate lesion activity [44,45].

Postoperative rehabilitation played a vital role in the patient’s recovery, particularly in restoring motor and cranial nerve function. Early initiation of neuromotor therapy has been shown to improve outcomes by leveraging the brainstem’s intrinsic capacity for plasticity. Studies highlight that intensive rehabilitation, including targeted physical and speech therapy, accelerates recovery and enhances long-term functionality. In our case, the patient’s significant improvement in mobility and swallowing underscores the importance of comprehensive postoperative care [46].

While significant advances have been made in the surgical management of brainstem cavernomas, key gaps remain. One of the most pressing challenges is the development of predictive models for risk stratification. Integrating genetic, clinical, and imaging data into machine learning algorithms could help identify patients who would benefit most from early intervention [47].

Technological innovations in neurosurgery, such as robotics-assisted microsurgery, augmented reality (AR), and virtual reality (VR), are poised to revolutionize the field [48]. These tools can enhance visualization and precision, particularly in deep-seated regions like the brainstem. Additionally, intraoperative imaging modalities, such as high-field MRI and fluorescence-guided resection, have the potential to further improve safety and completeness of resection [49].

Pharmaceutical therapy remains a largely untapped area in cavernoma management. Future research should focus on translating preclinical findings into clinical trials, with the goal of developing non-invasive treatments that stabilize lesions or prevent hemorrhages [50].

This case contributes to the growing body of evidence supporting early surgical intervention for symptomatic brainstem cavernomas. It reinforces the importance of tailoring surgical strategies to the lesion’s anatomy and prioritizing functional preservation over aggressive resection. Furthermore, it highlights the role of advanced imaging in guiding decision-making and ensuring postoperative stability.

The patient’s significant recovery, absence of major complications, and stable imaging findings demonstrate that even complex and high-risk lesions can be successfully managed with meticulous planning and execution. As technological and therapeutic advancements continue to evolve, cases like this one serve as a benchmark for achieving optimal outcomes in the management of rare and challenging neurovascular conditions.

## 4. Conclusions

This case underscores the intricate challenges of managing a giant pontine cavernoma and highlights the impact of tailored, timely intervention in restoring neurological function. The patient’s successful recovery demonstrates the potential for remarkable outcomes even in the context of life-threatening brainstem pathology. By carefully selecting the pterional transsylvian approach, the surgical team achieved significant decompression and stabilization while minimizing the risk of iatrogenic injury in this eloquent and anatomically dense region.

What makes this case particularly noteworthy is its emphasis on balancing the complexity of surgical resection with the importance of preserving functionality. The choice of subtotal resection allowed for effective decompression without compromising the surrounding critical tracts and nuclei, exemplifying a thoughtful approach that prioritizes long-term quality of life over aggressive lesion removal. This decision-making reflects a paradigm shift in modern neurosurgery, where precision and patient-centered care take precedence over maximal resection.

The role of imaging cannot be overstated in this case. Advanced diagnostic tools, such as SWI, were essential in defining the lesion’s hemorrhagic nature and guiding the urgency of intervention. Postoperative imaging further reinforced the success of the surgical strategy, confirming stability and the absence of rebleeding, which are crucial for preventing recurrence and ensuring sustained recovery.

This case also highlights areas for future exploration. The integration of advanced technologies, such as robotics-assisted microsurgery, augmented reality, and artificial intelligence in imaging, holds the potential to further refine the management of deep-seated cavernomas. Additionally, research into pharmacological therapies targeting vascular stabilization could open new avenues for non-invasive or adjunctive treatments, addressing the limitations of current surgical options.

Ultimately, this case contributes meaningful insights to the literature on brainstem cavernomas, demonstrating that even the most complex lesions can be managed effectively with thoughtful planning, technological integration, and multidisciplinary expertise. It serves as a reminder that precision, timing, and a patient-centered approach remain the cornerstones of success in neurovascular surgery.

## Figures and Tables

**Figure 1 jcm-14-02358-f001:**
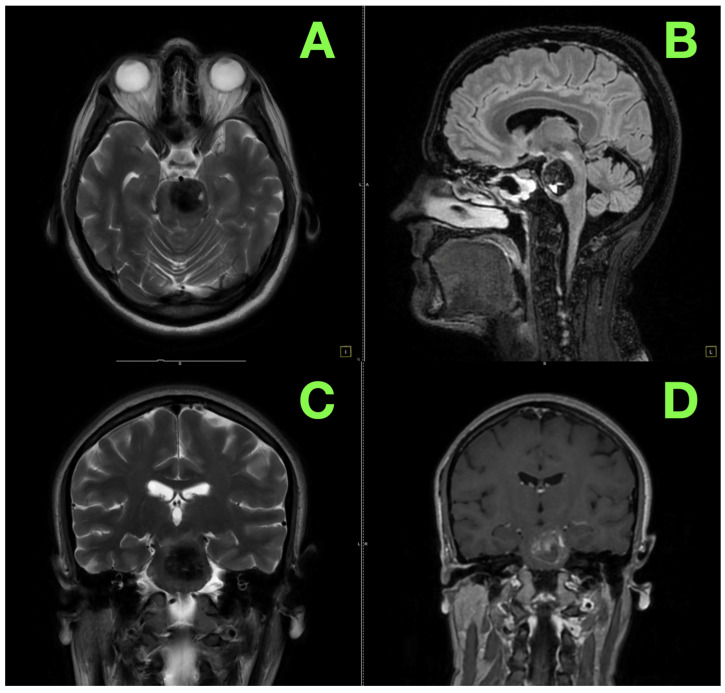
Preoperative MRI findings. (**A**): The lesion is seen prominently within the pons, exhibiting a heterogeneous signal. Hypointense regions correspond to hemosiderin deposition and chronic blood products, while mixed intensities suggest evolving hemorrhagic components. The lesion creates a mass effect, subtly displacing adjacent brainstem structures. (**B**): This view highlights the lesion’s vertical extent, delineating its interaction with the pontine anatomy. The compression of surrounding brainstem structures is evident, with partial effacement of the fourth ventricular floor and mild distortion of neighboring cisterns. (**C**): The lesion’s bilateral spread within the pons is clearly visualized, with asymmetric prominence on the left. The surrounding brainstem tissue appears mildly hyperintense, indicating reactive edema or secondary ischemia. (**D**): The lesion lacks significant enhancement, consistent with the avascular nature of cavernomas. Adjacent vascular structures, including the basilar artery, remain patent and unaffected.

**Figure 2 jcm-14-02358-f002:**
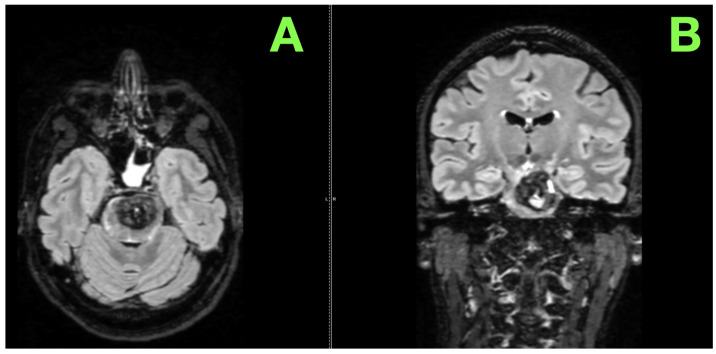
Advanced Imaging of the pontine lesion. (**A**): This sequence captures the lesion’s spatial distribution within the pons, showing prominent hyperintense signals at its periphery, likely reflecting reactive gliosis or perilesional edema. The central mixed signal suggests ongoing hemorrhagic activity, with regions of chronic blood products blending with subacute components. Notable mass effect is evident, compressing adjacent pontine structures and subtly effacing the fourth ventricular floor. (**B**): The lesion’s bilateral extension is highlighted, with an asymmetrical dominance on the left side. Hyperintense areas surrounding the lesion suggest secondary tissue reactions. The vertical orientation of the lesion relative to the brainstem axis is clearly delineated, emphasizing its anatomical disruption. The associated mass effect is visible, with subtle displacement of surrounding cerebrospinal fluid spaces.

**Figure 3 jcm-14-02358-f003:**
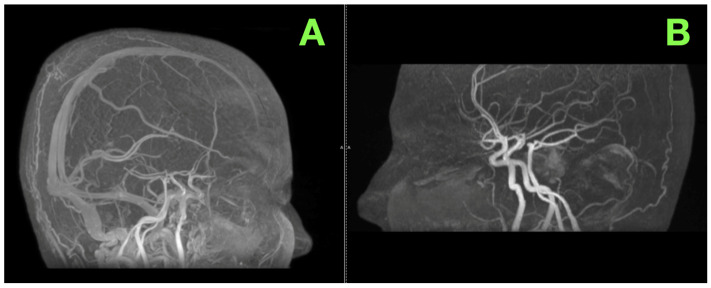
MRA of the intracranial vasculature. (**A**): Sagittal MRA reconstruction demonstrates the intact flow through the basilar artery and its branches, with no evidence of stenosis, aneurysms, or pathological vascular formations. The surrounding venous anatomy appears patent, with no signs of thrombosis or occlusion. (**B**): A lateral projection emphasizes the robust integrity of the vertebrobasilar system. The circle of Willis demonstrates no aneurysmal dilatations or aberrant configurations, providing confidence in the vascular stability during potential surgical intervention.

**Figure 4 jcm-14-02358-f004:**
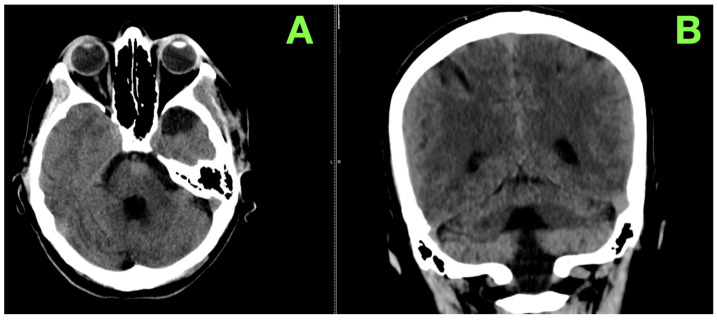
Three-month follow-up CT imaging. (**A**): This image demonstrates a residual iso- to hypodense area in the midline of the pons, consistent with postoperative scarring and stable sequelae of the subtotal resection. There is no evidence of recurrent hemorrhage, edema, or mass effect on the surrounding brainstem structures. (**B**): This view highlights the symmetric and midline alignment of the ventricular system, with no signs of hydrocephalus or compression of the fourth ventricle. The pterional craniotomy site is stable, showing no evidence of bone resorption, infection, or other complications.

**Table 1 jcm-14-02358-t001:** A comprehensive overview of key studies in the management of brainstem cavernomas, focusing on surgical techniques, treatment outcomes, imaging advancements, and long-term prognostic insights.

Author(s) and Year	Study Type	Population	Key Findings	Relevance to This Case
Gross et al. (2017) [33]	Retrospective cohort	168 patients with brainstem cavernomas	Early surgical intervention in symptomatic cavernomas reduces long-term morbidity; favorable outcomes in 78% of patients.	Supports the decision for early intervention in symptomatic cases, as in this patient with acute neurological deterioration.
Gui et al. (2019) [34]	Retrospective cohort	67 patients	Surgery improved functional outcomes in 85% of patients with hemorrhagic cavernomas; mortality rate of 1.3%.	Reinforces the benefits of surgical resection in hemorrhagic lesions while highlighting low perioperative mortality risks.
Ferroli et al. (2005) [35]	Case series	52 patients with brainstem cavernomas	Subtotal resection achieves excellent symptom relief while minimizing postoperative deficits in eloquent brainstem areas.	Validates the subtotal resection approach taken in this case to balance effective decompression and functional preservation.
Bulakci et al. (2012) [36]	Case Report	Focused on imaging modalities	SWI is critical for detecting hemosiderin deposits and hemorrhagic activity; fluid-fluid levels indicate active hemorrhage.	Highlights the role of advanced imaging, particularly SWI, in guiding surgical urgency and trajectory in this case.
Lawton et al. (2019) [37]	Single-center cohort	300 brainstem cavernomas	The pterional transsylvian approach provided safe access to ventral pontine lesions; good outcomes in 89% of cases.	Demonstrates the efficacy of the pterional approach used in this case to access and decompress a ventrally located lesion.
Abla et al. (2014) [38]	Multicenter review	300 patients with resected cavernomas	Risk of recurrence reduced with subtotal or gross resection; early intervention improves recovery.	Highlights the long-term stability achieved with subtotal resection and supports the surgical timing in this patient’s case.
Samii et al. (2001) [39]	Prospective study	101 brainstem cavernomas	Surgical outcomes improved with anatomical tailoring of approach; retrosigmoid best for dorsal lesions, pterional for ventral.	Supports the selection of the pterional route in this patient, optimizing access to the ventral pontine lesion.
Ramina et al. (2011) [40]	Retrospective cohort	43 patients with brainstem cavernomas	Microsurgical techniques (e.g., high-res microscopes) and neurophysiological monitoring reduce risks in brainstem surgery.	Reinforces the importance of precision and safety during resection in complex anatomical areas like the brainstem.
Mathiesen et al. (2003) [41]	Retrospective cohort	68 patients with brainstem lesions	Observation is appropriate for minimally symptomatic lesions; surgery reserved for recurrent or progressive symptoms.	Contrasts with the decision for urgent surgery in this case, highlighting the need for tailored management based on symptoms.
Falco et al. (2019) [9]	Monocentric series	34 cases analyzed	Emphasized role of multidisciplinary teams; perioperative care critical to reducing complications and enhancing outcomes.	Highlights the need for the integrated care provided in this patient’s surgical and postoperative management.

## Data Availability

The data presented in this study are available on request from the corresponding author.

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
