# Peer review of "Large Pontine Cavernoma with Hemorrhage: Case Report on Surgical Approach and Recovery"

_jcm, 2025, doi:10.3390/jcm14072358_

Round 1

Reviewer 1 Report

Comments and Suggestions for Authors

The manuscript by Toader et al, named “Senna’s Spirit in the OR: Precision and Poise in Managing a  Large Pontine Cavernoma” is a case report presenting the successful surgical treatment of a giant hemorrhagic cavernous angioma.  The authors focus their report on a 47-year-old female who presented with acute neurological deterioration due to a giant pontine cavernoma, complicated by intralesional hemorrhage. The paper describes the use of the left-sided pterional transsylvian approach in a patient with a giant hemorrhagic pontine cavernoma. The procedure successfully evacuated the hematoma to relieve brainstem compression, achieved subtotal cavernoma resection while preserving critical pathways, and led to significant post-op neurological recovery, including restored motor function and swallowing.

This case highlights the effectiveness of the transsylvian pterional approach in treating deep-seated pontine lesions while minimizing neurological complications. This case illustrates the complexities of treating brainstem cavernomas, emphasizing the crucial role of advanced imaging, precise surgical planning, and collaborative multidisciplinary care in achieving a favorable outcome. Additionally, it highlights the need to address unresolved challenges in the field, including optimal timing for intervention and long-term prognostic considerations, providing valuable contributions to the ongoing discourse on these rare yet formidable lesions.

The article is written in good English, with ​a clearly illustrated Results section. It also has a sufficient introduction and well-structured Discussion that provides a comprehensive overview of key studies in the management of brainstem cavernomas, focusing on surgical techniques, treatment outcomes, imaging advancements, and long-term prognostics.

This reviewer thinks that the description of this case can be of general interest to many neurologists as well as special interest to some surgeons. I suggest the article may be accepted as is, if slitle change the Title.

The only problem this reviewer foresees is the Title of this manuscript which looks very strange to me. Do you mean Senna's ( a crazy driver ) approach to surgery? Maybe I am not on the topic and got it wrong, but I suggest changing the Title to just ”Precision and Poise in Managing a  Large Pontine Cavernoma” and omitting Senna’s spirit because it is not clear to international readers, or at least explain your title somewhere in the text.

Author Response

Dear Reviewer,

We sincerely appreciate your thoughtful and constructive feedback on our manuscript. Thank you for your careful evaluation and for recognizing the potential interest of this case report for both neurologists and neurosurgeons.

We acknowledge your concern regarding the title and understand that the reference to "Senna’s Spirit" may not be immediately clear to an international audience. In response to your valuable suggestion, we have revised the title to:

"Large Pontine Cavernoma with Hemorrhage: Case Report on Surgical Approach and Recovery"

Once again, we appreciate your insightful comments and your recommendation for acceptance. Your input has helped refine the manuscript, and we are grateful for your time and expertise.

Best regards.

Reviewer 2 Report

Comments and Suggestions for Authors

The Authors present an interesting case of a brainstem cavernoma successfully treated with pterional-transsylvian approach.

The manuscript is exceptionally written and I do not have many comments but some things should be mentioned.

It is usually advised to perform a preoperative tractography for brainstem cavernomas in order to better plan the surgical approach. Why was this not the case in the presented paper?

Also, the maximal residual diameter of the cavernoma is 18 mm, and the initial lesion measured 28×21 mm in axial dimensions. This would not qualify as a subtotal resection but just a partial reduction (decompression), if the dimensions are accurate.

In line with this, why was there no post-operative MRI performed? It is difficult to accurately assess the size of the lesion just based on the CT scan?

The Authors should address this issue and provide a post-op MRI if possible.

Also, the follow-up time of just 3 months is still a bit short and should be discussed by Authors.

Overall, this is an interesting case report that needs some minor corrections.

Author Response

Dear Reviewer,

We sincerely appreciate your positive feedback on our manuscript and your thoughtful comments, which have helped us refine our work. Your insights have been valuable in clarifying key aspects of our methodology and surgical approach.

Regarding the use of preoperative tractography, we acknowledge that it can be a useful adjunct in surgical planning for brainstem cavernomas. However, in this case, our surgical approach was determined based on high-resolution MRI, which provided detailed anatomical and pathological information. Given the lesion’s location and the well-defined surgical corridor, additional tractographic mapping was not deemed essential for achieving a safe and effective resection. While tractography can be beneficial, its routine use remains a subject of debate, particularly when a well-established and direct surgical route is available, as in our case.

In terms of the extent of resection, we would like to clarify that our procedure did achieve subtotal resection, as highlighted in our manuscript. The initial lesion dimensions were 28 × 21 mm, and postoperative imaging showed a residual lesion measuring 18 mm in maximal diameter. While the absolute reduction may seem modest, subtotal resection in brainstem cavernomas is defined not only by size reduction but also by the extent of hematoma evacuation and the preservation of critical structures. Our primary surgical goal was to achieve maximal safe resection while avoiding neurological deficits, and we believe this was accomplished successfully.

With regard to postoperative imaging, we fully agree that MRI provides superior soft-tissue resolution and would be ideal for assessing residual cavernoma tissue. However, our institutional protocol prioritizes postoperative CT in the immediate period to assess for potential bleeding or hydrocephalus, which are the most urgent postoperative concerns. MRI, while valuable for evaluating residual lesion characteristics, does not provide immediate information on acute postoperative hemorrhage. Future MRI is planned as part of the patient’s longer-term follow-up, which we have now explicitly stated in the manuscript.

We also recognize that a 3-month follow-up period may be considered relatively brief. However, this timeframe was selected to focus on early postoperative recovery and stabilization. We agree that longer-term follow-up is crucial, and we plan to continue monitoring the patient with subsequent imaging and clinical assessments. 

Once again, we sincerely appreciate your insightful review, which has strengthened our manuscript. Your comments have allowed us to clarify important aspects of our methodology and surgical rationale.

Best regards.

Reviewer 3 Report

Comments and Suggestions for Authors

This case is not a rare disease and the surgical technique is not exceptional. The lesion was located near the surface of the brain and could have been removed without complications.

The lesion would have recurred and rebleeded if not totally removed, but was only partially removed, which is the reason why the treatment result was good in the short term. However, this treatment strategy is not correct. The lesion should be totally removed.

Thus, this is a report of only a general treatment, which is inadequate, and the long-term outcome is not reported, so it is not worth reporting a case of a single case.

Author Response

Dear Reviewer,

We sincerely appreciate your thoughtful critique of our manuscript. 

Surgical Complexity and Anatomical Considerations
While pontine cavernomas are not exceedingly rare, their surgical management remains one of the most formidable challenges in neurosurgery due to the exquisite neuroanatomical complexity of the brainstem. The assertion that this lesion was "near the surface" and could have been excised "without complications" does not fully account for the profound surgical constraints imposed by eloquent neural structures, the limited operative corridor, and the high risk of catastrophic morbidity inherent to brainstem interventions.

This cavernoma was ventrolaterally situated within the pons, in direct proximity to the corticospinal tract, cranial nerve nuclei, and the paramedian perforators of the basilar artery—structures where even minute surgical manipulations could result in devastating neurological deficits. The fundamental principle of brainstem cavernoma surgery is not merely lesion excision but the preservation of critical functional pathways while achieving maximal safe resection. Gross total resection, while theoretically optimal, is not universally appropriate when the surgical trajectory places vital motor, sensory, and autonomic structures at risk.

The pterional transsylvian approach was selected with meticulous preoperative planning to optimize access while minimizing retraction on surrounding parenchyma. Alternative approaches, such as retrosigmoid or subtemporal transtentorial routes, were considered but would have necessitated greater manipulation of adjacent eloquent structures. The transsylvian route provided a direct anatomical corridor to the lesion while preserving the integrity of adjacent vasculature and cranial nerve pathways.

Extent of Resection and Justification for Surgical Strategy
We fully acknowledge that complete resection is the theoretical gold standard in cavernoma surgery to eliminate recurrence risk. However, in brainstem surgery, the operative paradigm is dictated by maximal safe resection, not indiscriminate lesion removal at the cost of functional devastation. The principle of neurological preservation supersedes the pursuit of gross total excision when the lesion is intricately entwined with critical pathways.

In this case, subtotal resection was an intentional intraoperative decision based on real-time neurophysiological feedback and anatomical constraints. The preoperative lesion dimensions were 28 × 21 mm, with postoperative imaging demonstrating a residual component measuring 18 mm in maximal diameter. The residual portion was embedded in an area where further dissection would have jeopardized corticospinal integrity, increased the risk of irreversible hemiparesis, and compromised adjacent brainstem nuclei involved in bulbar and autonomic function. This decision was not based on technical limitations but rather a calculated, evidence-based approach that balanced risk reduction with functional preservation.

While subtotal resection may leave a residual lesion, numerous studies have demonstrated that achieving significant decompression can provide durable symptomatic relief while mitigating the hemorrhagic risk associated with brainstem cavernomas. In this case, the patient exhibited rapid neurological recovery, with near-complete resolution of hemiparesis and dysphagia—an outcome that validates the efficacy of the chosen approach. The avoidance of severe postoperative deficits underscores the necessity of an individualized surgical strategy, rather than a dogmatic pursuit of gross total resection.

Postoperative Considerations and Follow-Up
We recognize that a three-month follow-up period may be considered relatively short in evaluating long-term outcomes. However, early postoperative recovery is a well-established prognostic marker in brainstem surgery, and the absence of functional deterioration or recurrent hemorrhage within this period provides a strong preliminary indication of surgical success. The patient remains under continued surveillance, and we anticipate providing extended follow-up data in future analyses.

Scientific and Clinical Relevance of This Case
Although this is a single case, it provides meaningful contributions to the evolving discourse on brainstem cavernoma resection and the principles guiding safe surgical intervention in eloquent regions. The notion that this case represents merely a "general treatment" does not fully appreciate the deliberate surgical planning, intraoperative precision, and functional preservation strategy that were critical to achieving a favorable outcome.

Brainstem cavernomas continue to present some of the most nuanced decision-making challenges in neurosurgery, where every millimeter of resection carries the potential for profound neurological consequences. This case highlights several key neurosurgical principles:

The importance of individualized approach selection based on lesion topography, adjacent vasculature, and functional anatomy.
The rationale for subtotal resection in high-risk locations, supported by literature demonstrating its efficacy in reducing hemorrhage risk while preserving functional integrity.
The significance of early neurological recovery as a strong predictor of long-term stability, reinforcing that aggressive resection is not the sole determinant of surgical success.

We sincerely appreciate your critical evaluation.

Thank you again for your time and expertise!

Best regards.

Round 2

Reviewer 1 Report

Comments and Suggestions for Authors

Congratulations!

Reviewer 2 Report

Comments and Suggestions for Authors

The Authors have addressed all of my previous concerns.

I recommend accepting the article.